# Correlations between Mental Health, Physical Activity, and Body Composition in American College Students after the COVID-19 Pandemic Lockdown

**DOI:** 10.3390/ijerph20227045

**Published:** 2023-11-10

**Authors:** Luis Torres, Manuela C. Caciula, Alin S. Tomoiaga, Carmen Gugu-Gramatopol

**Affiliations:** 1Department of Exercise Science and Physical Education, Montclair State University, Montclair, NJ 07043, USA; 2Department of Health and Exercise Science, New Jersey City University, Jersey City, NJ 07305, USA; mcaciula@njcu.edu; 3Department of Accounting, Business, Analytics, CIS, and Law, Manhattan College, Bronx, NY 10471, USA; stomoiaga01@manhattan.edu; 4Department of Physical Education and Mountain Sports, Transilvania University of Brasov, 500036 Brasov, Romania; carmen.gg@unitbv.ro

**Keywords:** depression, anxiety, inactivity, sedentary, nutrition

## Abstract

Restrictions associated with the COVID-19 pandemic had forced American college students to significantly reduce their daily energy expenditure and increase their sedentary behaviors, thus presumably increasing mental health symptoms, decreasing physical activity levels, and enhancing the promotion of unhealthy eating habits. This study aimed to explore the correlations between mental health symptoms, physical activity levels, and body composition in college students in the years following the pandemic, focusing on the lingering effects of lockdown measures. American college students completed pre-existing, well-validated surveys for both mental health (Hospital Anxiety and Depression Scale) and physical activity (International Physical Activity Questionnaire—Long Form). Body composition was assessed in person with the use of an Inbody 570 device. Of the 90 American college students (*M* age = 22.52 ± 4.54, 50 females) who participated in this study, depressive and anxious symptom scores consistent with heightened symptomatology were reported by 58% of the participants (*N* = 52), moderate borderline symptomatology by 17% (*N* = 15), and asymptomatology by 25% (*N* = 23). In regard to physical activity, 79% (*N* = 71) of the students were highly physically active, 18% (*N* = 16) were moderately active, and 3% (*N* = 3) reported low levels of physical activity. Additionally, 46% (*N* = 41) of the students maintained an unhealthy body fat percentage based on the World Health Organization recommendations. Strong, significant relationships were found between anxiety and depression symptomatology and body fat percentage (positive correlation, *p* = 0.003) and between anxiety and depression symptomatology and skeletal muscle mass (negative correlation, *p* = 0.015), with said symptomatology increasing with added body fat and decreasing with added skeletal muscle mass. The COVID-19 pandemic lockdown restrictions affected American college students through an increase in mental health symptomatology and a deterioration in overall body composition.

## 1. Introduction

The SARS-CoV-2 (severe acute respiratory syndrome coronavirus 2), more commonly referred to as COVID-19, has wreaked havoc on all facets of higher education since its inception in late 2019. The associated COVID-19 pandemic lockdown restrictions that emerged as a direct result of the spread of this virus in the United States remained primarily in place from 13 March 2020 (the date on which the United States declared a National Emergency concerning the COVID-19 outbreak and began enforcing subsequent lockdown restrictions) up until 13 May 2021 (the date on which the Centers for Disease Control and Prevention updated their safety guidelines to allow fully vaccinated individuals to participate in indoor and outdoor activities, large or small, without wearing a mask or physical distancing) [1]. Some of these lockdown restrictions across the country included, but were not limited to, stay-at-home orders, travel restrictions, social distancing measures, and isolation and quarantine requirements [2,3,4]. During this time, over 25 million American college students were affected by these pandemic lockdown restrictions as many universities across the country sent residential and commuter students home, canceled in-person campus activities, and transitioned to online instruction [5,6]. As a result, students reduced their daily energy expenditure and increased their sedentary behaviors, thus presumably increasing mental health symptoms, decreasing physical activity levels, and enhancing the promotion of unhealthy eating habits. Evidence as to how these variables may be correlated in subsequent years after the lockdown restrictions, however, remains minimal or non-existent.

It is crucial to explore the contemporary relationships among mental health symptoms, physical activity levels, and body composition of American college students, given the pronounced impact of the COVID-19 pandemic lockdown on these variables. Early studies have supported the notion that the pandemic lockdown has significantly worsened the onset of mental health symptoms, such as anxious symptoms, in college students [7,8,9]. Depressive symptoms are also worth investigating as the results of previous well-known epidemiological studies have demonstrated that depression and anxiety co-occur frequently in college students despite the fact that they are different mental health disorders [10,11]. The pandemic lockdown undoubtedly brought on significant stressors and disruptions in daily life along an uneasy timeline without a definitive end [6]. The impact of the pandemic lockdown restrictions on the mental health of American college students cannot be understated, especially given the fact that inadequate support networks and insufficient coping skills commonly found within adolescents and young adults are already an existing source of psychological distress [12].

Furthermore, natural disasters and pandemics have long been associated with a significant restriction of overall physical movement [5]. Researchers have supported this statement with their finding showing that the COVID-19 pandemic lockdown has had a negative impact on the physical activity levels of American college students through a decrease in overall physical activity levels and an increase in sedentary behaviors [13,14]. Factors such as social isolation, decreased social support, lack of intrinsic motivation, and limited access to exercise facilities have contributed to these happenings [15]. It is reasonable to surmise that the noted impact of the COVID-19 pandemic lockdown on the physical activity levels and sedentary behaviors of American college students correlates with an overall decrease in lean body mass and an increase in body fat percentage, thus affecting body composition [13,14]. Despite the fact that there is currently limited research on body composition changes during the COVID-19 pandemic lockdown, some researchers have focused on self-reported changes in daily eating habits. American college students have reported increased unhealthy snacking and emotional eating during the pandemic lockdown, as well as decreased consumption of fruits and vegetables [16,17]. Given the timely presented evidence, it is possible that there are strong relationships among mental health symptoms, physical activity levels, and body composition of American college students after the COVID-19 pandemic lockdown. A better understanding of these relationships in the United States will assist clinicians in developing multifaceted interventions that will help address these issues in this population. The purpose of this study was to determine the correlations between mental health symptoms, physical activity levels, and body composition of American college students after the COVID-19 pandemic lockdown in order to better inform the development of future health-improving interventions in this population. In the context of this study, it was primarily hypothesized that relationships between mental health, physical activity, and body composition would be found; secondly, it was also hypothesized that certain physical activity and body composition variables could be used to predict mental health symptomatology.

## 2. Materials and Methods

### 2.1. Design and Participants

In the current cross-sectional study, we tested a total of 90 undergraduate students enrolled at a four-year public university located in New Jersey, United States. It is pertinent to note that the university’s overall undergraduate enrollment for the 2022–2023 academic year was approximately 4500 students. This study was advertised to undergraduate and graduate students of all majors through mass emails, recruitment flyers, and personal in-class pleas over a six-month period from November 2022 to April 2023. Once students agreed to participate in this study, they were invited to an Exercise Science Lab for the provision of an informed consent form. Once written consent was obtained, each participant was asked to complete a QR-code-linked demographic survey via Google Forms, in which information pertinent to age, biological sex, ethnicity, and current academic year was gathered. After the completion of the demographic survey, each participant was provided with the Hospital Anxiety and Depression Scale (HADS) and the International Physical Activity Questionnaire (IAPQ) to collect data on self-reported mental health symptoms and self-reported physical activity levels, respectively. The HADS was delivered through another Google Forms survey, while the IPAQ was delivered on paper, for the purposes of facilitating participant interpretation and data analysis.

Lastly, each participant was asked to remove their shoes and socks for a bioelectrical impedance body composition assessment using an Inbody 570 device. All participants were thanked for their time and provided with a complimentary individualized Inbody 570 informative handout at the conclusion of the study. Each data collection session lasted approximately 30 minutes. Additional information on all utilized instruments will be provided in subsequent sections. All researchers in this study completed the Collaborative Institutional Training Initiative (CITI) program training for research involving human subjects before beginning participant recruitment. All collected data were de-identified, kept confidential, and stored in a password-encrypted Google Drive account. The IBM SPSS 27.0 Statistical Package was used to analyze all collected data at the end of the data collection period.

### 2.2. Instruments and Variables

The Hospital Anxiety and Depression Scale (HADS) is a 14-item questionnaire designed to measure symptoms of depression and anxiety [18]. The HADS consists of two subscales that are composed of seven questions for symptoms of depression (HADS-D) and seven questions for symptoms of anxiety (HADS-A). Each item on the questionnaire contains responses that are individually scored on a scale from 0 to 3, with higher scores indicating a higher level of symptom frequency. The combined scores of emotional distress range from 0 to 42, with scores of 11 or higher indicating a clinically significant mood disorder. Specifically, scores that range from 0 to 8, from 9 to 11, and 11 or higher coincide with normal, borderline, and abnormal HADS categories, respectively. The sensitivity and specificity of approximately 0.80 for both the HADS-A and HADS-D are very similar to the sensitivity and specificity achieved by the General Health Questionnaire (GHQ); correlations between the HADS and other commonly used mental health questionnaires, such as the Beck Depression Inventory (BDI), the Clinical Anxiety Scale, and Spielberger’s State-Trait Anxiety Inventory (STAI), are in the range of 0.49 to 0.83 [19]. De Munter et al. [20] used the HADS to study symptoms of depression and anxiety after injury in a study involving 4239 adult trauma patients admitted into a hospital between August 2015 and December 2016. Fischerauer et al. [21] used the HADS to measure symptoms of depression and anxiety in a study examining fear avoidance to physical function and pain intensity in 102 injured athletes. Bjelland et al. [19] found that the HADS performed well in assessing symptom severity and caseness for depression and anxiety in both somatic, psychiatric, and primary patients and in the general population in a literature review of 747 studies.

Self-reported physical activity was assessed using the International Physical Activity Questionnaire—Long Form (IPAQ). Developed by Craig et al. [22], the IPAQ-LF is a 27-item questionnaire that addresses five activity domains: job-related physical activity; transportation physical activity; housework, house maintenance, and caring for family; recreation, sport, and leisure-time physical activity; and time spent sitting. Within the context of each domain, participants are instructed to think about all the time that they spent on vigorous-intensity activities, moderate-intensity activities, and walking within the last seven days through a series of example-guided questions. There are two forms of output from scoring the IPAQ. Results can be reported in categories (low activity levels, moderate activity levels, or high activity levels) or as a continuous variable (MET minutes/week). Using the instrument’s scoring protocol, total weekly physical activity is estimated by weighing the time spent in each activity intensity with its estimated metabolic equivalent (MET) energy expenditure. The IPAQ scoring protocol assigns 3.3 METs to walking, 4.0 METs to moderate-intensity activity, and 8.0 METs to vigorous-intensity activity. Those who score high on the IPAQ engage in either vigorous-intensity activity on at least 3 days, achieving a minimum total physical activity of at least 1500 MET minutes a week, or 7 or more days of any combination of walking, moderate-intensity, or vigorous-intensity activities, achieving a minimum total physical activity of at least 3000 MET minutes a week. Scoring a moderate level of physical activity on the IPAQ means participants are performing some activities that are likely to be equivalent to half an hour of at least moderate-intensity physical activity on most days. Those who score moderate on the IPAQ engage in either 3 or more days of vigorous-intensity activity and/or walking of at least 30 min per day, 5 or more days of moderate-intensity activity and/or walking of at least 30 min per day, or 5 or more days of any combination of walking, moderate-intensity, or vigorous-intensity activities, achieving a minimum total physical activity of at least 600 MET minutes a week. Scoring a low level of physical activity on the IPAQ means that participants are not meeting any of the criteria for either moderate or high levels of physical activity [23].

Kim et al. [24] found an adequate convergent validity of the I-PAQ (corrected mean effect size = ESp) with other instruments for walking, total moderate-intensity physical activity, vigorous-intensity physical activity, and total physical activity (ESp = 0.32, 0.45, 0.49, and 0.39, respectively). An excellent concurrent validity of the I-PAQ was identified when looking at the correlation of time spent in vigorous-intensity physical activity compared to accelerometer monitoring (q = 0.71, *p* < 0.001) [25]. Also, strong positive relationships were observed between accelerometer-based activity monitor data and IPAQ data for total PA (rho = 0.55, *p* < 0.001), but a weaker relationship for moderate PA (rho = 0.21, *p* = 0.051) [26]. Furthermore, Craig et al. [22] showed that the IPAQ produced repeatable data (Spearman’s rho clustered around 0.8), with comparable data obtained from the short and long forms.

The InBody 570, a bioelectrical impedance analysis (BIA) device, was used to estimate body composition in this study, with a particular emphasis on body fat percentage and skeletal muscle mass. BIA is widely utilized in the fitness industry, clinical settings, and research due to its non-invasiveness, ease of use, and relatively low cost compared to other methods like DEXA (Dual-Energy X-ray Absorptiometry) [27,28,29]. Typically, body composition analysis involves determining the percentages of fat mass and fat-free mass in the body. Fat-free mass includes skeletal muscle, organs, bones, and water. Miller et al. [27] showed that multi frequency BIA used by the InBody 570 scale is a valid method for determining body fat percentage when compared to DXA (r = 0.94, *p* < 0.0001). Also, the InBody 570 device has been found to provide consistent and reproducible results when used under standardized conditions. However, it is important to note that certain factors, such as hydration status, food intake, and physical activity, can influence the results of any BIA-based method [30]. Therefore, all participants in this study were instructed via email upon scheduling their testing sessions to follow the recommended InBody preparatory steps, such as avoiding exercise and food intake for at least three hours prior to testing, avoiding the intake of alcohol and caffeine for at least 24 h, and refraining from using the shower or sauna before the test. Participants were encouraged to maintain their normal fluid intake the day before testing, to remove shoes and socks, and to use the restroom before testing.

### 2.3. Data Analysis

An a priori power analysis was conducted using G*Power version 3.1.9.7 [31] to determine the minimum sample size required to test our regression model. The results indicated the required sample size to achieve 80% power for detecting an effect of 0.40 at a significance criterion of α = 0.05 was *N* = 88. Thus, the obtained sample size of *N* = 90 was adequate to test all study hypotheses. A nonparametric Spearman correlation test was run to look for relationships among the variables of HADS scores, weekly METs, body fat percentage, and SKMM. A linear regression analysis was used to examine the impact of body fat percentage, skeletal muscle mass, weekly physical activity, sex, ethnicity, and academic year on mental health scores. The linear regression assumptions of linearity, independence, homoscedasticity, and normality of residuals were tested using both graphical methods and hypothesis testing (multicollinearity was tested as well).

## 3. Results

A total of 90 undergraduate American college students (*M* age = 22.52 ± 4.54, 40 males, 50 females) participated in this study; the ethnicities and academic years of the participants are presented in Table 1. The participants reported a mean depressive and anxiety symptom score of 13.37 ± 6.8, with the overall scores being consistent with heightened symptomatology reported by 58% of the participants (*N* = 52), while 17% (*N* = 15) showed moderate borderline symptomatology, and 25% (*N* = 23) showed normal symptomatology. In regard to physical activity, the participants reported 10,673.34 ± 9564.47 weekly METs, with 79% (*N* = 71) of them being highly physically active, 18% (*N* = 16) being moderately active, and 3% (*N* = 3) reporting low levels of physical activity. In the context of body composition, male participants displayed a body fat percentage of 17.00 ± 8.66% and SKMM of 34.9 ± 6.73 kg, and female participants displayed a body fat percentage of 32.9 ± 9.93% and SKMM of 25.5 ± 4.30 kg.

Significant relationships were found between anxiety and depression symptomatology and body fat percentage and skeletal muscle mass, with said symptomatology increasing with added body fat (rho = 0.29, *p* = 0.004) and decreasing with added skeletal muscle mass (rho = −0.26, *p* = 0.012). There were very low, insignificant correlations found between weekly METs and HADS scores, between weekly METs and body fat percentage, and between weekly METs and SKMM. To better understand these relationships, a linear regression analysis model was also employed to examine the impact of body fat percentage, SKMM, weekly METs, sex, ethnicity, and academic year on HADS score (F-statistic = 2.261, *p* = 0.04529). The results indicated that, again, body fat percentage had a significant positive relationship with HADS score at the 0.05 level (Beta = 0.1964, *p* = 0.01345), suggesting that as body fat percentage increased, HADS score also increased. Additionally, SKMM showed an almost significant negative relationship with HADS score (Beta = −0.2559, *p* = 0.05297), implying that an increase in SKMM was associated with a decrease in HADS score. Every percentage increase in body fat correlated with an average HADS score increase of 0.16 (see Figure 1), while every kilogram increase in SKMM correlated with an average score decrease of 0.24 (see Figure 2). The other predictors (i.e., weekly METs, sex, ethnicity, and academic year) did not significantly contribute to the regression model. The regression model accounted for approximately 14.05% of the variance in HADS score (multiple R^2^ = 0.1405), which reduced to 7.835% when adjusted for the number of predictors (adjusted R^2^ = 0.07835). This suggested that while these predictors provided good insight into which factors might influence HADS score, a significant amount of variance remained unexplained.

## 4. Discussion

This study aimed to explore the correlations between mental health symptoms, physical activity levels, and body composition in college students in the years following the pandemic, focusing on the lingering effects of lockdown measures. Mental health conditions, such as depression and anxiety, are salient in the lives of adolescents and young adults as these individuals experience, and work through, a variety of life events, such as leaving home, establishing independence from parents or guardians, dating and beginning romantic relationships, and possibly entering or continuing their studies in an institution of higher education [12]. As noted in this study, the latter event was further complicated with the enforcement of COVID-19 pandemic lockdown restrictions in the United States. The finding of this study that 58% of the participants reported a heightened depressive and anxious symptomatology illustrated that the COVID-19 pandemic lockdown might have had a negative relationship with the mental health of American college students. The physical activity levels among American college students varied widely before the COVID-19 pandemic lockdown. Some studies have shown that a significant portion of college students did not meet the recommended guidelines for physical activity [31]. The *Physical Activity Guidelines for Americans, 2nd edition,* suggests that individuals should aim for at least 150 min of moderate-intensity aerobic activities or 75 min of vigorous-intensity activities per week, along with muscle-strengthening activities twice a week [32]. During the COVID-19 pandemic lockdown, when many colleges implemented remote learning and social distancing measures, physical activity patterns were likely affected. Some students might have experienced a decrease in physical activity due to limited access to recreational facilities, closure of gyms, reduced opportunities for organized sports, and increased sedentary behaviors associated with remote learning [14]. However, the results of this study showed that 79% of the students reported high levels of physical activity, 18% reported moderate levels, and 3% reported low levels of physical activity. These findings could indicate that the COVID-19 pandemic lockdown influenced the physical activity habits of American college students differently, and that individuals’ self-reported physical activity levels varied widely based on personal factors, motivations, and circumstances. Some college students might have found alternative ways to engage in physical activity, such as home workouts, outdoor activities, online fitness classes, or virtual challenges. These findings are somehow similar to the findings reported by Lopez-Vaneciano et al. [33], which suggest that those students who met the current minimum PA recommendations before the lockdown generally met the recommendations also during pandemic-related confinements.

The physiological variables of interest in this study were body fat percentage (%BF) and skeletal muscle mass (SKMM). %BF is one of various indicators used to assess overall health and fitness, with excessive body fat often being associated with an increased risk of various health conditions, including cardiovascular disease, type 2 diabetes, and certain cancers [34]. Also, carrying excess body fat is known to place additional stress on the joints, especially the knees, hips, and ankles which, over time, can contribute to joint pain, osteoarthritis, and limited mobility [35]. Excess body fat has also been shown to restrict lung expansion and decrease lung capacity, leading to decreased respiratory function and contributing to conditions like sleep apnea, which can have negative effects on overall health [36]. Similar to %BF, SKMM plays a crucial role in overall health for several reasons, including metabolic health and regulation, physical function and performance, bone health, and aging and longevity [37]. More specifically, a higher proportion of skeletal muscle mass has been shown to increase the basal metabolic rate, leading to greater energy expenditure and helping to maintain a healthy body weight [38]. Also, having adequate muscle mass improves strength, endurance, balance, coordination, and overall bone health, which results in ease of performing daily living activities and reduced risks of fractures and osteoporosis [39]. Furthermore, skeletal muscle acts as a storage site for glucose and glycogen and helps regulate the blood sugar levels by absorbing glucose from the bloodstream. This is known to be extremely important for individuals with diabetes and insulin resistance [40]. Lastly, maintaining skeletal muscle mass has been shown to be increasingly important with aging. Sarcopenia, an age-related loss of muscle mass and strength, can lead to frailty, functional decline, and an increased risk of falls and disabilities. By maintaining or increasing muscle mass through exercise and resistance training, the effects of aging can be mitigated, resulting in a longer and healthier lifespan [41,42].

The COVID-19 pandemic lockdown led to changes in routines and increased sedentary behaviors for many students [43]. Limited access to gyms, reduced participation in organized sports, and potential changes in dietary habits due to stress, isolation, or disruptions in daily routines might have contributed to weight gain, increased body fat percentage, and decreased skeletal muscle mass for some students [43]. Similar findings were reported in collegiate athletes by Cholewinski et al. [44], with COVID-19 lockdown leading to a significant decrease in fat-free mass and increases in fat mass and body fat percentage. There are no accepted or standardized norms for skeletal muscle mass (in kg) specifically for college students, partly because muscle mass can vary greatly depending on a range of factors, including age, sex, height, physical activity level, and genetics [45]. However, some approximate general ranges based on sex and age for adult population have been proposed in the National Health and Nutrition Examination Survey (NHANES), with males (age 20–39) being recommended to have between 33.0 and 39.9 kg of skeletal muscle mass and females (age 20–39) between 21.0 and 25.9 kg. While college students in this study displayed acceptable SKMM values, the heightened %BF indicates the need for nutritional support and interventions specifically designed for American college students.

Given the relationships found between mental health symptoms and body compositions among American college students in this study, future health-improving interventions should be targeted to address these two main components. College students have shown a strong willingness to accept and adhere to interventions delivered via smartphone applications for psychological well-being; these interventions are often focused on cognitive behavioral therapy techniques that seek to reduce stress, anxiety, depression, and risky behaviors such as alcohol and tobacco uses, and improve knowledge on sexual activity [46]. At this point in time, the majority of these applications primarily focus on prevention (e.g., Nod, BioBase, Thrive: Feel Stress Free, Headspace, and Destressify) rather than on the provision of human-supported treatment through licensed therapists and coaches; only four existing applications currently offer human-supported treatment organized specifically for only college students (i.e., Lantern, TAO: Therapy Assisted Online, StudiCare Stress, and Mind the Moment) [46,47]. This notion gives rise to the hypothesis as to whether American college students, who have routinely displayed a heightened depressive and anxious symptomatology, may better benefit from the use of psychological well-being applications that directly connect them to specific counseling and psychological services provided by their respective institutions. Alternatively, smartphone diet-tracking applications, such as FatSecret, MyPlate, and MyFitnessPal, have proven to be effective in helping American college students lose weight and understand their dietary habits; however, many of these applications often do not consider the importance of providing targeted advice to reduce anxious and depressive characteristics while working toward overall body composition goals [48]. The design of future applications should be encompassing of all strategies that seek to improve the mental health and body composition of American college students, regardless of self-reported physical activity levels. Furthermore, these future applications should be specifically tailored to reflect the specific resources and services that an institution can provide for its college students. The authors of this study will seek to create and pilot these types of applications for college students at several different institutions in a future study.

In the future, researchers may also want to replicate studies similar to this one with a larger sample size, thus improving the predicted effect size of the obtained results. It is worth noting that the general limitations of this study included the subjective nature of the assessment of self-reported physical activity levels, the potential failure of participants in not strictly following the highly encouraged pre-testing body composition instructions, variation when testing occurred throughout the day, and not controlling for the enrolled academic programs of the participants. Although the I-PAQ is a well-validated questionnaire in assessing self-reported levels of physical activity, there are other more accurate instruments, such as accelerometers.

## 5. Conclusions

The COVID-19 pandemic lockdown restrictions affected American college students, who showed an increase in mental health symptomatology and a deterioration in overall body composition. The findings gathered in this study from college students provided evidence of an existing positive relationship between mental health and body fat percentage and a negative relationship between mental health and skeletal muscle mass. It was further found that these same body composition variables could perhaps be used to predict symptoms of depression and anxiety. Collectively, it is recommended that future health interventions for American college students should focus on strategies to reduce stress, anxiety, and depressive characteristics, as well as provision of nutritional information on healthy eating, regardless of self-reported physical activity levels. Future research revolving around the creation and utilization of comprehensive smartphone applications to promote such strategies and provide nutritional information, while connecting college students to the counseling and psychological or nutritional services available on their respective campuses, should be especially considered in order to move forward in the post-COVID-19 pandemic lockdown restriction era in the United States.

## Figures and Tables

**Figure 1 ijerph-20-07045-f001:**
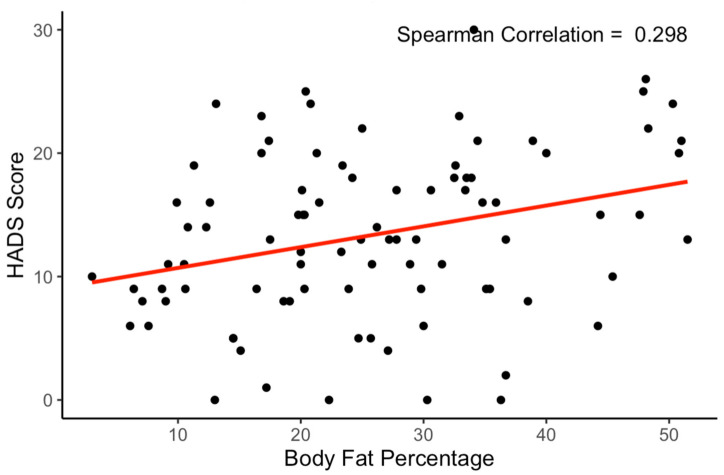
Correlation between body fat percentage and HADS score. *Note. N* = 90. Every percentage increase in body fat correlates with an average HADS score increase of 0.16 (*p* = 0.0044).

**Figure 2 ijerph-20-07045-f002:**
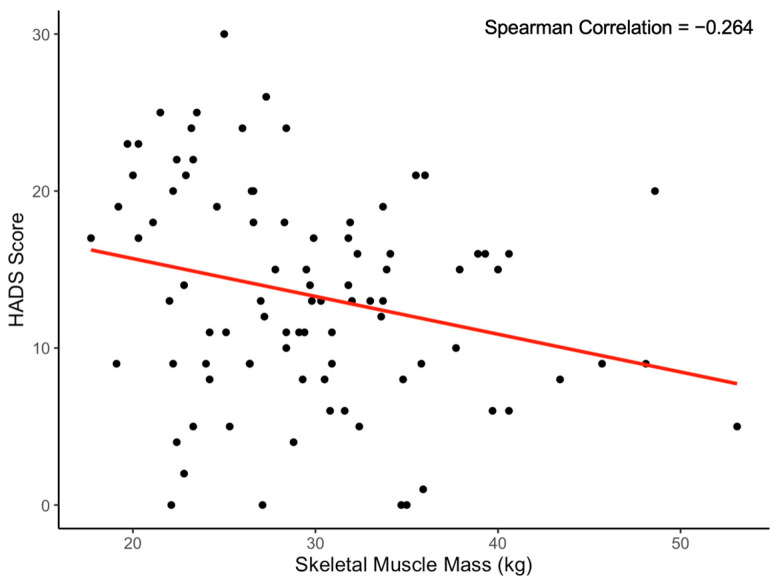
Correlation between skeletal muscle mass and HADS score. *Note. N* = 90. Every kilogram increase in skeletal muscle mass correlates with an average HADS score decrease of 0.24 (*p* = 0.012).

**Table 1 ijerph-20-07045-t001:** Demographic characteristics of participants.

Characteristic	*N*	%	HADS Mean	HADS SD	Weekly METs Mean	Weekly METs SD	BF% Mean	BF% SD	SKMM (kg) Mean	SKMM (kg) SD
Sex										
Male	40	44.4%	11.95	6.10	9607.20	8087.221	17.0	8.66	34.9	6.73
Female	50	55.5%	14.5	7.17	11,526.26	10,603.308	32.9	9.93	25.5	4.30
Race/Ethnicity										
Asian	8	8.9%	15.4	6.5	10,068	4968	26.9	13.8	28.2	7.34
Black/African	16	17.8%	15.19	7.24	14,741	15,231	29.54	13.16	29.38	8.11
Hispanic/Latino	41	45.6%	12.85	6.88	8748.37	7811.76	26.26	11.71	28.99	5.57
White/Caucasian	17	18.9%	13.06	6.24	12,102.38	9029.72	23.85	13.29	30.66	7.44
Other	8	8.9%	11.00	7.39	9972.62	6387.07	19.40	8.24	33.19	11.74
Academic Level										
Freshman	10	11.1%	12.6	8.06	10,435.85	5856.18	23.62	9.23	27.77	5.36
Sophomore	9	10.0%	15.44	7.32	7777.67	6192.42	27.12	10.95	28.63	10.24
Junior	22	24.4%	12.55	6.29	11,759.93	11,965.88	21.55	11.59	31.79	7.00
Senior	49	54.5%	13.51	6.79	10,762.14	9592.84	27.97	13.00	29.31	6.97

## Data Availability

The data presented in this study are available from the corresponding author upon request. The data are not publicly available in an effort to properly safeguard college student data gathered from an American public university.

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
