# Peer review of "Correlations between Mental Health, Physical Activity, and Body Composition in American College Students after the COVID-19 Pandemic Lockdown"

_ijerph, 2023, doi:10.3390/ijerph20227045_

Round 1
Reviewer 1 Report
Comments and Suggestions for Authors
The manuscript presents intriguing findings pertaining to the relationships between the measured variables (namely, mental health, body composition, and physical activity) within the study's participants. However, there is a pressing need for significant enhancements in multiple facets of the manuscript, including its structural arrangement, logical coherence, the robustness of its evidence, and the alignment between the reported results and the drawn conclusions.
Introduction
Lines 56-60: It is recommended that the study's purpose be relocated to the conclusion of the introduction, just before the main research hypothesis (lines 94-98).
Lines 56-60: In the study's objectives, the authors employ the terms "current correlation" and "after the pandemic," which require further elucidation. In general, issues such as the study's temporal framework, baseline information, and comparators demand clarification, as it appears that significant contextual details are absent. This issue is pervasive throughout the entire manuscript, notably in the methodology and the interpretation of results/discussion.
Line 66: Consider augmenting your references [7-9] with more contemporary publications, such as the following: https://pubmed.ncbi.nlm.nih.gov/37761690/
Lines 69-70: Proper citation is essential.
Materials and Methods
Line 106: Specify whether the consent was obtained in written or electronic form.
Lines 106-117: Provide information about the average time required for procedure completion.
Line 136: Enhance the clarity of scale names, e.g., "Scale for Depression and Anxiety Symptoms" (similar actions should be taken for the subheading titles in 2.3 and 2.4).
Lines 209-214: Clarify when and how participants received these instructions (e.g., via email or other means).
Lines 224-238: Consider condensing this section; the current content could be more suitably placed in the introduction or discussion sections.
Results
Lines 240-250: Tabulate the scores of the scales per demographic group in Table 1.
Discussion
Lines 307-311: The discussion around the "increasing pattern of present anxious and depressive symptomatology" lacks clarity. Reference 41 indicates the impact of various stressors on students' mental health but does not directly compare or associate with the point made by the authors. What is conspicuously absent here is a reference to provide context for their observations. Are they comparing the mental health status before the pandemic, such as the baseline in the Zivin et al. paper? In such a case, a more recent publication would be pertinent. Alternatively, are they comparing the students to the general population?
Lines 318-332: Improve the logical connections and conclusions to better convey the central concept of diversity concerning the effects on students' physical activity.
Lines 342-348: Consider utilizing one of the muscle mass indices instead of the mass of muscular tissue. For instance, refer to https://www.ncbi.nlm.nih.gov/pmc/articles/PMC6382481/.
Lines 363-388: Significantly shorten this section. The reference to the development of applications and their role could be a concise paragraph highlighting the study's prospects. In its current form, it disrupts the balance between the study's purpose and the actual research question. Additionally, discuss the study's limitations.
Comments on the Quality of English Language
No specific comments
Reviewer 2 Report
Comments and Suggestions for Authors
This is an important topic in public health covering the correlation between Mental Health, Physical Activity, and Body Composition in American College Students. Please see my comments below:
Line 27, please clarify if the correlation was positive or negative for each variable
Line 52, there was no recommendation or enforcement on reducing daily energy expenditure, please rephrase or provide appropriate references.
Line 84, please provide appropriate citations.
Line 133, please define a medium effect.
What was the primary outcome/s of this study?
Line 216, to 238 should be moved to the discussion section.
Line 251, what was the unhealthy body fat percentage? Please put the numbers.
Line 253 and 254, belong to the method section.
Table 2 is not needed, the information has been stated in the text
Line 294, this study was a cross-sectional study testing correlations, not an RCT, the conclusion of a causal effect is inappropriate.
Line 334, needs a reference
Line 339, please specify in your results, what were the healthy norms for each gender and what were the means for your population.
The conclusion is long and repetitive. Please merge it with the discussion
Comments on the Quality of English Language
Some grammatical errors detected. Please proofread the text.
Reviewer 3 Report
Comments and Suggestions for Authors
Dear authors,
You can find the comments of my review in the attached document.
Kind regards

Round 2
Reviewer 1 Report
Comments and Suggestions for Authors
The authors have provided a significantly improved version of their manuscript. Major issues previously have now been properly addressed.
Reviewer 2 Report
Comments and Suggestions for Authors
The quality of the manuscript has been improved.
Reviewer 3 Report
Comments and Suggestions for Authors
Dear authors,
I would like to congratulate you on the improvements you have made to the manuscript to improve its scientific quality.
Best regards